# Maintaining the Quality of Mechanical Thrombectomy after Acute Ischemic Stroke in COVID(-)19 Patients

**DOI:** 10.3390/brainsci12111431

**Published:** 2022-10-25

**Authors:** Yu-Hao Chang, Nien-Chen Liao, Yuang-Seng Tsuei, Wen-Hsieh Chen, Chiung-Chyi Shen, Yi-Chin Yang, Chung-Hsin Lee

**Affiliations:** 1Department of Neurosurgery, Neurological Institute, Taichung Veterans General Hospital, Taichung 407219, Taiwan; 2Institute of Clinical Medicine, National Yang-Ming University, Taipei 112304, Taiwan; 3Department of Critical Care Medicine, Taichung Veterans General Hospital, Taichung 407219, Taiwan; 4Institute of Medicine, Chung Shan Medical University, Taichung 40201, Taiwan; 5Department of Post-Baccalaureate Medicine, College of Medicine, National Chung Hsing University, Taichung 40227, Taiwan; 6Department of Neurosurgery, Neurological Institute, Tri-Service General Hospital, Taipei 114202, Taiwan; 7Department of Radiology, Taichung Veterans General Hospital, Taichung 407219, Taiwan; 8Department of Industrial Engineering and Enterprise Information, Tunghai University, Taichung 407224, Taiwan; 9Department of Medical Imaging and Radiological Science, Central Taiwan University of Science and Technology, Taichung 406053, Taiwan

**Keywords:** acute ischemic stroke, COVID-19, mechanical thrombectomy

## Abstract

The COVID-19 pandemic has become increasingly worse worldwide since it was discovered in China in late December 2019. Easy contact transmission between people and a low to moderate mortality rate may cause failure in medical health services if there is no proper personal protective equipment for personnel. During the pandemic, patients with acute ischemic stroke with large-vessel occlusion who required immediate treatment through mechanical thrombectomy (MT) were still being sent to the emergency room. Knowing how to maintain effective treatment standards has become our concern. We used a retrospective, single-center study to select COVID-19 (-) patients with acute ischemic stroke undergoing mechanical thrombectomy during the years 2020–2021. Patients with acute ischemic stroke with large-vessel occlusion received mechanical thrombectomy were compared with patients admitted from December 2020 to May 2021 (the pre-COVID-19 group) and those from June 2021 to November 2021 (the during COVID-19 group). Furthermore, the time disparity of mechanical thrombectomy was compared between these two groups. Of patients confirmed with acute ischemic stroke (AIS) with large-vessel occlusion (LVO) during the study period, 62 were included. Compared with the pre-COVID-19 group (34 patients; median age, 70.5 years), the during COVID-19 group (28 patients; median age, 71.5 years) showed no major median time difference in door-to-computed-tomography-angiography (CTA) time (19.0 min vs. 20.0 min, *p* = 0.398) and no major median time difference in door-to-groin-puncture time (118.0 min vs. 109.0 min, *p* = 0.281). In our study, with a prepared protocol for the pandemic having been established in the healthcare system, we could see no difference between the pre-pandemic and during-pandemic time periods when using mechanical thrombectomy to treat COVID-19 (-) patients of AIS with LVO. By means of a quick-PCR test during triage, there was no time delay to perform MT or any lowering of safety protocol for workers in the healthcare system.

## 1. Introduction

In late December 2019, a previously unknown coronavirus, now named the 2019 novel coronavirus, emerged from Wuhan, China, resulting in serious breakouts throughout many cities in China before spreading globally. The disease was then officially named Coronavirus Disease-2019 (COVID-19 by the WHO on 11 February 2020) [1,2,3,4]. COVID-19 is a potential zoonotic disease with a low to moderate (estimated 2–5%) mortality rate [5]. Person-to-person transmission may occur through droplets or contact transmission, and if there is a lack of stringent infection control or no proper personal protective equipment available, the virus may endanger first-line healthcare workers. The estimated proportion of asymptomatic COVID-19-positive patients is believed to be between 17.9 and 30.8%, with 74% of these asymptomatic carriers being contagious [6]. Taiwan, an island nation of approximately 24 million people, has been extraordinarily successful in preventing a major calamity during the epidemic. As of March 2021, there had been only 10 deaths and slightly more than 1000 infected cases. Most people generally lived without fear of becoming infected with SARS-CoV-2. However, Taiwan’s government did eventually announce a lockdown, the third emergency alert of the pandemic on 19 March 2021, due to COVID-19 spreading from the national airport to the general population [7,8].

Acute ischemia stroke is one of the most significant causes of morbidity and mortality worldwide. Mechanical thrombectomy (MT) has established its role as the standard of treatment in patients with acute ischemic stroke with large-vessel occlusions (LVOs), depending on appropriate patient selection and timely reperfusion [9]. The procedure is necessary for both saving the brain and decreasing neurologic deficits [10,11] and is performed as soon as possible once acute ischemic stroke has been diagnosed [12,13]. During the pandemic, for each patient who required an acute thrombectomy after being given a possible diagnosis and going through a special control channel, all would be equipped with a full set of isolation equipment [14]. This would not only increase medical costs and treatment times, it would also, at the same time, under the full set of isolation equipment, cause the quality of treatment from the thrombectomy physician to be greatly reduced [15].

The COVID-19 viral RNA reverse-transcription polymerase chain reaction (RT-PCR) test is used to rapidly screen patients who require immediate admission to the hospital [16,17,18,19]. When a patient is diagnosed in the emergency room and requires an immediate endovascular mechanical thrombectomy, rapid nucleic acid detection is performed. The aim of this study was to evaluate patients of AIS with LVO who required MT, and any association with clinical outcomes amongst those patients both before and during the COVID-19 pandemic in the central region of Taiwan. This was done by means of reverse-transcription polymerase chain reaction (RT-PCR) to screen the COVID-19 (-) patients of AIS with large-vessel occlusion who required an immediate mechanical thrombectomy. We compared the door-to-CTA time and the door-to-puncture time between the pre-COVID-19 and the during-COVID-19 groups in MT for AIS.

## 2. Materials and Methods

### 2.1. Patient Population

In this retrospective, single-center study, we summarize our experience with patients of AIS with LVO who underwent mechanical thrombectomy for acute ischemic stroke during the years 2020–2021. The inclusion criteria were as follows: (1) acute ischemic stroke due to LVO, including the internal carotid artery (ICA), the middle cerebral artery (the MCA M1 and M2 segments), the basilar artery (BA), and the intracranial segment of the vertebral artery (VA), as confirmed by computer tomography angiography (CTA) and treated with mechanical thrombectomy; (2) the time from symptom onset to reperfusion, within 6 h for anterior circulation, within 24 h for posterior circulation; and (3) the 6 ≤ baseline National Institute of Health Stroke Scale (NIHSS) score ≤ 30.

Data, including demographics, the baseline NIHSS, the initial laboratory results, comorbidities, and pre-admission medications, were collected from all patients. Cerebral baseline imaging was performed on multislice computed tomography (non-contrast CT and CT-angiography) to confirm acute cerebral ischemia. According to the Guidelines of the Taiwan Stroke Society for the Management of Patients with Ischemic Stroke [20], intravenous thrombolysis (rt-PA) (0.9 mg/kg, with 10% as a bolus) was to be administered if patients arrived in the window time of <4.5 h and had no contraindication prior to mechanical thrombectomy. This study was approved by the Institutional Review Board of Taichung Veterans General Hospital, Taichung, Taiwan (approval code: CE21506A).

### 2.2. Acute Ischemic Stroke Registry

The following parameters of baseline clinical characteristics were collected: gender, age, and pre-stroke modified Rankin Scale (mRS) score (scores on the modified Rankin Scale range from 0 to 6, with 0 indicating no symptoms; 1, symptoms without clinical disability; 2, slight disability; 3, moderate disability; 4, moderately severe disability; 5, severe disability; and 6, death). The baseline and follow-up after MT National Institutes of Health Stroke Scale (NIHSS) score; the sites of occlusion vessels, including the internal carotid artery, the middle cerebral artery M1 segment, or basilar artery occlusion; the time intervals (door-to-imaging time and door-to-groin puncture time); the treatment profile; and the functional outcomes evaluated by the mRS at 3 months were all obtained. Moreover, patient medical history, including hypertension, dyslipidemia, diabetes mellitus, ischemic stroke, and atrial fibrillation, were also mentioned. Occlusion sites were determined using CT angiography at the emergency room. The time of symptom onset was defined as either the time when symptoms emerged or when the patient last felt normal if the time of symptom onset was not seen.

### 2.3. Statistical Analysis

In the present research, the data of each assessment were collected to avoid uncertainty, with the results presented as mean ± SD. The Mann–Whitney U test was used to determine the differences between groups. A *p* value of less than 0.05 was considered to be statistically significant.

### 2.4. Acute Ischemic Stroke Workflow

All of the patients presenting to the emergency room are triaged immediately on the basis of the presence or absence of COVID-19 symptoms, which is where they will receive the advisable workup (Figure 1). Simultaneously, patients presented with symptoms of acute stroke must be evaluated rapidly for the purpose of instant reperfusion therapies where applicable. With regards to this reason, during the pandemic we have made it a practice that all patients referred to the stroke team are evaluated and receive reperfusion therapy (if needed) as soon as possible, with any or suspected cases of COVID-19 (+) throughout the brain-attack pathway being isolated until the workup results are released. To minimize exposure, one on-call neurologist (a stroke fellow or resident neurologist) will attend to the patient. The donning of appropriate PPE based on the hospital’s infection control center (ICC) has already been well-rehearsed. If presenting with large-vessel occlusion, the patient is referred to a neurointerventionist and moved to the angiography suite/catheterization room for MT. Radiologic technicians in the angiography suite are pre-alerted if the patient is to undergo endovascular intervention. Acute-stroke patients continue to be presented to the emergency department but may not display the usual symptoms of COVID-19 infection. The stroke-team response, and stroke-team management, must be performed within the shortest possible time in order to minimize the worsening of functional outcomes without compromising the safety of the medical team.

### 2.5. Mechanical Thrombectomy

All mechanical thrombectomy technique procedures were performed by neurointerventionalists certified by the Interventional Neuroradiology Department of Neuroradiological Society of Taiwan. MT was performed under general anesthesia. MT was performed using catheter aspiration or a stent retriever.

## 3. Results

A total of 62 patients (33 females, 29 males) (median age, 71.5 years; median NIHSS score of 18) with a diagnosis of AIS with LVO in the study period, who underwent mechanical thrombectomy, were included. Of these, 34 patients were allocated to the pre-COVID-19 group, while 28 were allocated to the during COVID-19 group. All patients were COVID-19-negative. The baseline characteristics and clinical outcomes of patients are shown in Table 1. The time delay in mechanical thrombectomy is shown in Table 1 and Figure 2.

Compared with the pre-COVID-19 group, the during COVID-19 group showed no major difference in median door to CTA time (19 min vs. 20 min, respectively; *p* = 0.398). Additionally, there was no major difference in median door-to-puncture time (Figure 3A) (118.0 min vs. 109.0 min, respectively; *p* = 0.281) and no different rate of reaching an intent DTP time of ≤90 min (15.2% vs. 17.9%, respectively; *p* = 1.000) (Figure 3B) and 120 min (54.5% vs. 67.9% respectively; *p* = 0.452) (Figure 3C). The prognosis of the during-COVID-19 group showed no difference in the rate of TICI score = 2b\2c\3 (30% vs. 21%) and no difference in the rate of mRS score after MT three months ≤2 score (pre-COVID-19 group/during COVID-19 group: 21.4%:27.3%) (Figure 4).

However, in the central area of Taiwan where the prevalence of COVID-19 was low, there was less efficiency for each AIS patient who required MT to undergo the protection route. In our hospital, the only tertiary national medical hospital in central Taiwan, COVID-19 viral RNA Reverse-transcription polymerase chain reaction (RT-PCR) is used to rapidly screen patients who need to prepare for MT. If the RT-PCR result was negative (PCR (-)), the patient was taken to the angiography room via the standard way (Figure 5A), while if the patient displayed a RT-PCR positive result (PRC (+)) or no PCR result, they were taken to an isolated angiography room for MT via an outdoor isolated way (Figure 5B).

In Taiwan, we had been previously made aware of the proper procedures during the SARS outbreak years ago and have thus proven ourselves to be self-reliant and in alignment with international healthcare organizations while fighting the COVID-19 virus [21]. We presented the results of the influence of the COVID-19 pandemic on acute stroke care in this tertiary hospital by comparing the results of MTs performed during the half year period prior to the COVID-19 pandemic lockdown in Taiwan to those of the half year period during lockdown in Taiwan. Thus, our data reveal the impact the virus had on acute stroke care during the epidemic’s first wave peak. Patients of AIS with LVO must be evaluated for MT as soon as possible. In our hospital, we have a responsibility to establish a strong ability to fulfill the necessary requirements for performing the procedure. Prior to the onset of the COVID-19 pandemic, we had set up the proper protocols and corrected any issues that may have surrounded the treatment of AIS patients during our monthly meetings, which were held in association with discussing the stroke (Figure 1).

## 4. Discussion

Taiwan has established four major principles—rapid measures, early deployment, prudent action, and transparency, in response to the pandemic while also creating the ‘Taiwan Model’ for dealing with it [22]. When new respiratory contagious diseases occur in a widespread manner, such as the outbreak of COVID-19, healthcare workers’ beliefs regarding infection prevention and control guidelines (IPC) become even more important. Healthcare workers have indicated that there are several factors that influence their ability and confidence when following IPC guidelines during the course of managing respiratory infectious diseases. These factors are connected to the guidelines themselves and how they are communicated, including support from managers, workplace culture, training, physical space, access to and trust in personal protective equipment (PPE), and a willingness to deliver sound patient care [23].

Donning proper PPE since the declaration of the COVID-19 pandemic while also ensuring the safety of our medical team while delivering timely treatment has certainly been a challenge. Personal protective equipment can potentiate heat stress, which in turn may have a negative impact on the wearer’s performance, safety, and well-being [15]. Additionally, the WHO has sent warnings that the acute shortage in the global supply of PPE has been caused by both climbing requirement and snaping up, and this improper use is putting lives at risk due to COVID-19. At last, although health workers who participated in the treatment of AIS patients are at risk of infection and require proper PPE, they are taught to use PPE in the right place and avoid any inappropriate consumption [24]. The PCR test at the triage in our acute ischemic stroke workflow may not only protect the MT team without wearing PPE routinely from COVID-19 infection but also maintain the quality of the COVID-19 (-) patients of AIS with LVO receiving MT.

The dominant finding of this study is that no decrease in the number of acute stroke with LVO-receiving MT patients was seen. There is every likelihood that the result basically connected to the fact that Taiwan, an island nation of approximately 24 million people, has been extraordinarily successful in containing COVID-19 [8]. Other studies have already reported differences, attracting unprecedented attention to acute stroke. Most of them emphasize a significant decline in the number of acute-stroke patients presenting during the pandemic’s peak [25,26,27,28,29], as well as what has been seen as a light influence in regions where the confirmed number of COVID-19 cases was lower amongst the population [30]. The drop in the number of acute-stroke patients admitted has been more obvious in older patients. The elderly are afraid of catching COVID-19 owing to the generalized information that increasing age or comorbidity could lead to poorer outcomes regarding the treatment of the infection. However, social distancing may also have played a considerable role during the lockdown. Isolation may complicate the ongoing care of these patients and may delay the onset of any stroke symptoms being noticed by their families [26,28].

Yoshimoto T reported that prolonged time in mechanical thrombectomy treatment was found during the COVID-19 period in comparison with the pre-COVID-19 period. Nevertheless, they noticed that there was a reduction in both the door-to-needle time and the door-to-groin puncture time in the later period of the COVID-19 [31]. However, our results show that there was no delay in reperfusion procedure. This is because simple workflow improvements for streamlining in hospital triage and performing critical workup at transferring hospitals can produce reductions in the door-to-puncture time [32]. We had successfully accessed our AIS patients due to having set up workflow procedures prior to the pandemic. Even during the pandemic, the regular workflow was censured by the stroke team, with the quick PCR test effectively being added to the workflow at the triage station of the ER without causing any time-delay. Consequently, we were able to maintain solid results for MT in COVID-19 (-) patients of AIS with LVO during the pandemic.

It is widely known that good clinical outcomes after MT for AIS depend strongly on time [32,33,34]. Since it is necessary to rescue time for the patients of AIS and treatment usefulness is basically depending on timely access to the needed therapy, the present guidelines suggest that the evaluation of reperfusion therapy should be processed quickly, thus preventing any unnecessary and potentially time-consuming procedures [35,36,37,38]. Seung Hwan Kim reported that NIs achieved a shorter door-to-puncture time than non-NI neurologists. The door-to-puncture time of their NIs group revealed 135.2 ± 50.0 min [39]. Our stroke team has already set up an instant telecommunication application and an acute ischemic stroke workflow, which could save each step time of the process, such as shortening the waiting time of PCR test, image reading, and team communication. Neurointerventionists receive information regarding the image of CTA with perfusion scan immediately, just as patients with AIS at ER have finished their CTA study. Our door-to-puncture time result revealed no major difference before (mean: 118 min) or during the COVID-19 pandemic (mean: 109 min).

There were several limitations in our study. Firstly, our study is an observational single-center study design. Some selection bias could have occurred. In the COVID-19 pandemic, emergency transfer processes are largely influenced by pre-hospital emergency medical technicians. Secondly, the sample amount of this study was relatively small. The third limitation is that we had no comparison between the number of MT performed in the study and the potential number of candidates. Because this is a retrospective study, the information of pre-hospital patients was not well collected. Stroke occurrence could had remained undetected in time during lockdown. We will make sure to discuss this for further prospective investigations with large sample sizes.

## 5. Conclusions

In our study, with a prepared protocol for the pandemic having already been put in place in the healthcare system, we can see no major difference between the pre-pandemic and during-pandemic periods, when mechanical thrombectomy was being used to treat COVID-19 (-) patients of AIS with LVO. By means of a quick PCR test, there was no time delay in performing MT, and the safety of the members of the healthcare team was maintained.

## Figures and Tables

**Figure 1 brainsci-12-01431-f001:**
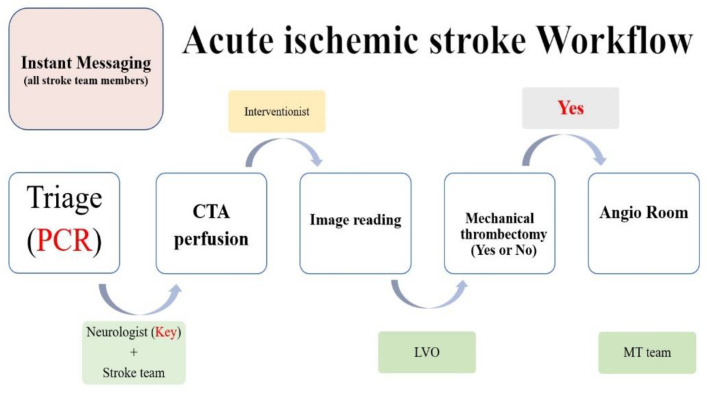
Prior to the COVID-19 pandemic, we developed protocols and corrected the shortcomings of treating AIS patients during monthly stroke-related meetings. (CTA: Computed Tomography Angiography. LVO: Large Vessel Occlusion. MT: Mechanical Thrombectomy.)

**Figure 2 brainsci-12-01431-f002:**
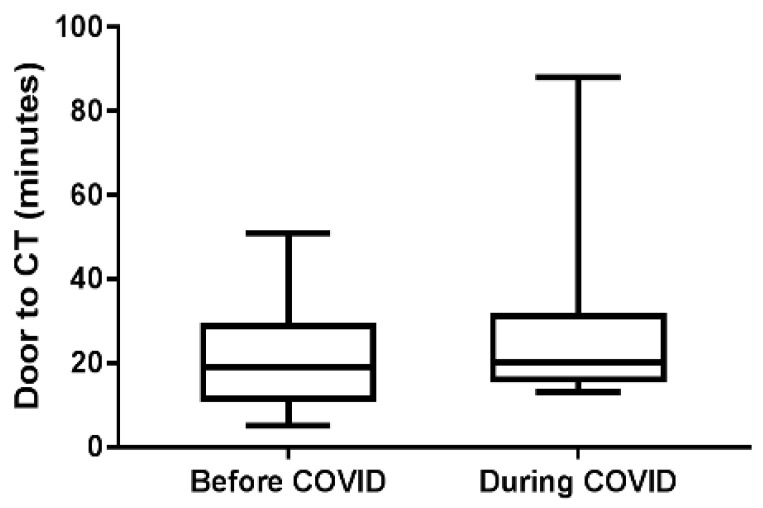
A comparison of door-to-CT times between the during COVID-19 and pre-COVID-19 period groups.

**Figure 3 brainsci-12-01431-f003:**
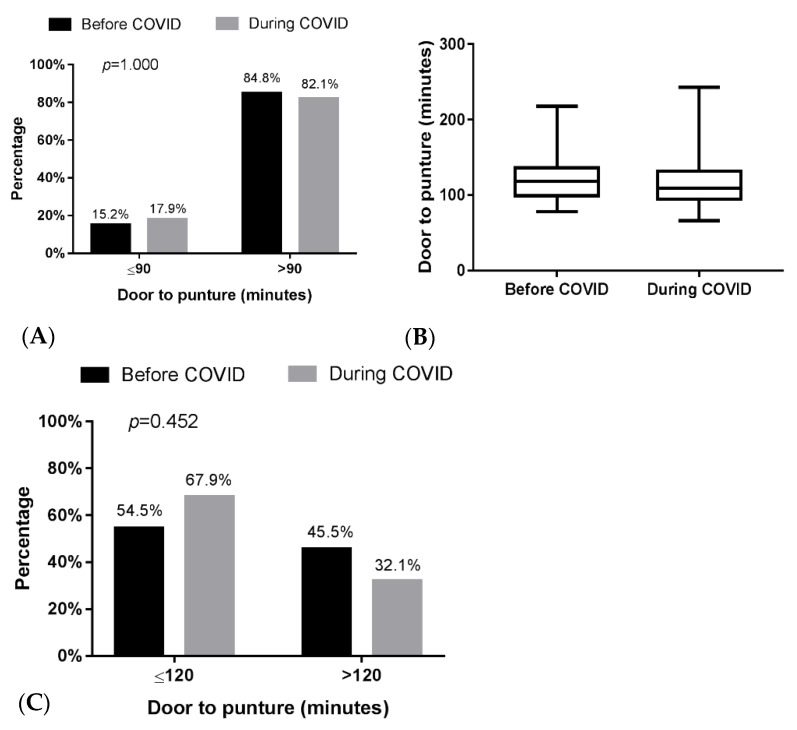
Comparison of door-to-puncture times between the during and pre-groups. (**A**) The door-to-puncture times compared. (**B**) The door-to-puncture time of 90 min compared. (**C**) The door-to-puncture time of 120 min compared.

**Figure 4 brainsci-12-01431-f004:**
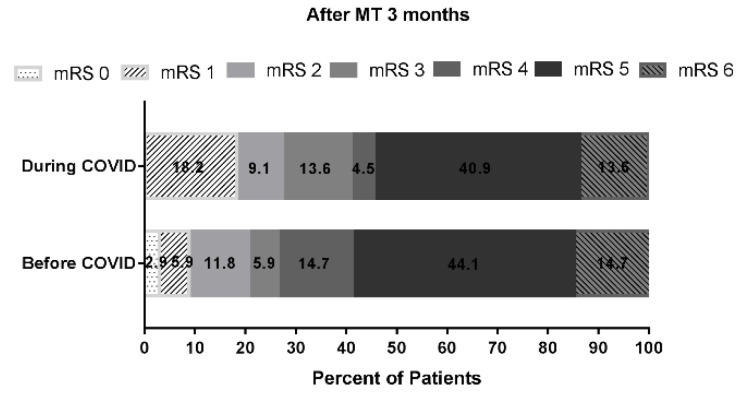
Functional outcomes at 3 months from onset according to the Modified Rankin Scale Score. During COVID (*n* = 28); pre-COVID (*n* = 34). Favorable outcome as defined by a modified Rankin Scale (mRS) score less than or equal to 2 at 3-month follow-up. The rate of mRS score after MT three months ≤ 2 score presented that pre-COVID-19 group was 21.4%, compared to 27.3% during COVID-19 group.

**Figure 5 brainsci-12-01431-f005:**
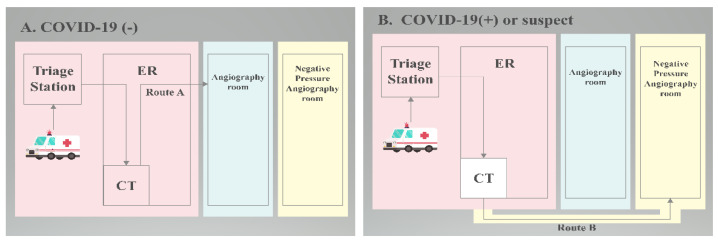
Patients suspected of acute ischemic stroke were immediately tested for COVID-19 PCR at the triage station in the emergency department and then sent to the CT room. Flow chart: (**A**) regular return visits. (**B**) Exceptional circumstances require isolation procedures. (ER: Emergency Room. CT: Computed Tomography.)

**Table 1 brainsci-12-01431-t001:** The baseline characteristics and clinical outcomes of patients.

	All (*n* = 62)		Before COVID (*n* = 34)		During COVID (*n* = 28)		*p* Value
**Age**	71.5	(64.75, 80.25)	70.5	(63.8, 79.3)	71.5	(67.0, 82.0)	0.276
**Female**	33	(53.2%)	15	(44.1%)	18	(64.3%)	0.184
**Location**							0.494
anterior circulation	53	(85.5%)	28	(82.4%)	25	(89.3%)	
posterior circulation	9	(14.5%)	6	(17.6%)	3	(10.7%)	
**Medical history**							
hypertension	47	(75.8%)	27	(79.4%)	20	(71.4%)	0.665
hyperlipidemia	19	(30.6%)	13	(38.2%)	6	(21.4%)	0.249
diabetes mellitus	22	(35.5%)	16	(47.1%)	6	(21.4%)	0.067
atrial fibrillation	32	(51.6%)	18	(52.9%)	14	(50.0%)	1.000
ischemia heart disease	19	(30.6%)	11	(32.4%)	8	(28.6%)	0.964
NIHSS at ER	18.0	(12.0, 24.0)	17.0	(11.5, 22.0)	18.0	(15.3, 27.8)	0.143
Door to CT, minutes	20.0	(15.0, 29.0)	19.0	(11.5, 28.7)	20.0	(16.0, 31.0)	0.398
Door to punture, minutes	114.0	(97.0, 134.0)	118.0	(99.0, 136.0)	109.0	(94.8, 131.8)	0.281
**TICI Score**							0.654
0	2	(3.3%)	1	(2.9%)	1	(3.7%)	
1	3	(4.9%)	1	(2.9%)	2	(7.4%)	
2a	5	(8.2%)	2	(5.9%)	3	(11.1%)	
2b	13	(21.3%)	9	(26.5%)	4	(14.8%)	
2c	10	(16.4%)	7	(20.6%)	3	(11.1%)	
3	28	(45.9%)	14	(41.2%)	14	(51.9%)	
2b\2c\3	51	(83.6%)	30	(88.2%)	21	(77.8%)	
**Symptomatic Hemorrhage in 36HR**	7	(13.0%)	3	(10.7%)	4	(15.4%)	0.699
**mRS 3 m 0–2**	13	(23.2%)	7	(20.6%)	6	(27.30%)	0.799
**mRS 3 m 6**	8	(14.3%)	5	(14.7%)	3	(13.6%)	1.000

Demographic (*n* = 62) Mann–Whitney U test, Median (IQR).

## Data Availability

Not applicable.

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
