# Peer review of "Maintaining the Quality of Mechanical Thrombectomy after Acute Ischemic Stroke in COVID(-)19 Patients"

_brainsci, 2022, doi:10.3390/brainsci12111431_

Round 1

Reviewer 1 Report

This is an interesting study on the use of MT during COVID pandemic in Taiwan. It is known to be very organized due to previous SARS emergencies. The data even if on a small population may be  interesting for the medical community. here my remarks since the manuscript needs to be improved.

abstract: CTA spell out first time

abstract: specify location of the study and type (single centre) in the abstract (readers often look just at abstract)

Introduction: please correct and clarify since the sentence  in the introduction stating that "its role as a form of first-line treatment  for acute ischemic stroke due to large vessel occlusions (LVOs). " is not fully supported by current guidelines that  state how "MT is the standard of care in patients with LVO-related acute stroke. Appropriate patient selection and timely reperfusion are crucial. Further randomized trials are needed to inform clinical decision making with regard to the mothership and drip-and-ship approaches, anesthaesia modalities during MT, and to determine whether MT is beneficial in patients with low stroke severity or large infarct volume."  

ref: Turc G, Bhogal P, Fischer U, et al. European Stroke Organisation (ESO) – European Society for Minimally Invasive Neurological Therapy (ESMINT) Guidelines on Mechanical Thrombectomy in Acute Ischaemic StrokeEndorsed by Stroke Alliance for Europe (SAFE). European Stroke Journal. 2019;4(1):6-12. doi:10.1177/2396987319832140

methods: respectfully for the Polish medical community the citation in the manuscript of the "Guidelines of the Polish Neurological Society for the Management of Patients with Ischemic Stroke" seems not in line with mainstream international trend the reference 6 is not readable for the public.

results: please provide overall data on the general population admitted for stroke in 2020-2021 and excluded from MT

discussion: PPE need to spell out at first time

page 7 from line 197  to 223 you describe the organization of the model. Usually the discussion is dealing with the results of the study the organization is not matter of discussion in this case. I suggest to move this in the method section. You can speculate in the discussion section of different organizational models and confront them with yours.

page7 line 214 Personal Protection Equipment instead of PPE once using the acronym just do it at the first time 

Discussion: please clarify since usually when a technique is efficient we observe a rise in the use of it year by year. In the next years a rise in number of MT may be observed in Taiwan. COVID may reduce the expected slope rise and slow down the diffusion of the technique. The number of potential candidates to MT are usually below 5-10 % of ischemic strokes admitted and we have no data on the total number of stroke admitted hospitals participating to this study. The number of MT performed in the study is consistent with the potential number of candidates ? there is still a gap to fill ? Readers would be interested to know also this data.

Author Response

Response to Reviewer 1 Comments

This is an interesting study on the use of MT during COVID pandemic in Taiwan. It is known to be very organized due to previous SARS emergencies. The data even if on a small population may be interesting for the medical community. Here my remarks since the manuscript needs to be improved.

Point 1: abstract: CTA spell out first time

Response 1: abstract (line39) CTA was been corrected to Computed Tomography Angiography.

Point 2: abstract: specify location of the study and type (single centre) in the abstract (readers often look just at abstract)

Response 2: We corrected the sentence and added the information on abstract (line31) as below:

We used retrospective, single-center study to select COVID-19 (-) patients with acute ischemic stroke undergoing mechanical thrombectomy during the years 2020–2021.

Point 3: Introduction: please correct and clarify since the sentence  in the introduction stating that "its role as a form of first-line treatment  for acute ischemic stroke due to large vessel occlusions (LVOs). " is not fully supported by current guidelines that  state how "MT is the standard of care in patients with LVO-related acute stroke. Appropriate patient selection and timely reperfusion are crucial. Further randomized trials are needed to inform clinical decision making with regard to the mothership and drip-and-ship approaches, anesthaesia modalities during MT, and to determine whether MT is beneficial in patients with low stroke severity or large infarct volume."  

ref: Turc G, Bhogal P, Fischer U, et al. European Stroke Organisation (ESO) – European Society for Minimally Invasive Neurological Therapy (ESMINT) Guidelines on Mechanical Thrombectomy in Acute Ischaemic StrokeEndorsed by Stroke Alliance for Europe (SAFE). European Stroke Journal. 2019;4(1):6-12. doi:10.1177/2396987319832140

Response 3: As the reviewers recommendation, We deleted the sentence(line67)” Mechanical thrombectomy (MT) has established its role as a form of first-line treatment for acute ischemic stroke due to large vessel occlusions (LVOs)”. We added the new sentence” Mechanical thrombectomy(MT) has established its role as the standard of treatment in patients of acute ischemic stroke with large vessel occlusions(LVOs), depending on appropriate patient selection and timely reperfusion.

And, we add reference 37. (Turc G, Bhogal P, Fischer U, et al. European Stroke Organisation (ESO) – European Society for Minimally Invasive Neurological Therapy (ESMINT) Guidelines on Mechanical Thrombectomy in Acute Ischaemic StrokeEndorsed by Stroke Alliance for Europe (SAFE). European Stroke Journal. 2019;4(1):6-12. doi:10.1177/2396987319832140)

Point 4: methods: respectfully for the Polish medical community the citation in the manuscript of the "Guidelines of the Polish Neurological Society for the Management of Patients with Ischemic Stroke" seems not in line with mainstream international trend the reference 6 is not readable for the public.

Response 4: As the reviewers recommendation, we corrected the wrong mention as below(line104, line371). We delected the sentence of the Guidelines of the Polish Neurological Society for the Management of Patients with Ischemic Stroke, and added the Guidelines of Taiwan Stroke Scoiety for the Management of Patients with Ischemic Stroke.

And we attached the reference 38 (Chen CH, Hsieh HC, Sung SF, et al. 2019 Taiwan Stroke Society Guideline for intravenous thrombolysis in acute ischemic stroke patients. Formos J Stroke 2019;1:1-22.)

Point 5: results: please provide overall data on the general population admitted for stroke in 2020-2021 and excluded from MT

Response 5: As the reviewers recommendation, We(the Stroke Center of Taichung Veterans General Hospital, Taichung, Taiwan) collected the total 534 cases of pateints of acute ischemic stroke admitted to our hospital in 2020-2021.

Point 6: discussion: PPE need to spell out at first time

Response 6: As the reviewers recommendation, We had corrected it on line 226 (personal protective equipment (PPE)).

Point 7: page 7 from line 197  to 223 you describe the organization of the model. Usually the discussion is dealing with the results of the study the organization is not matter of discussion in this case. I suggest to move this in the method section. You can speculate in the discussion section of different organizational models and confront them with yours.

Response 7: As the reviewers recommendation, we very agreed that the organization of the model should be moved to the method section (line 132). And we have discussed the issue in line 255.

Point 8: page7 line 214 Personal Protection Equipment instead of PPE once using the acronym just do it at the first time 

Response 8: As the reviewers recommendation, we had corrected it(page7 line 226).  

Point 9: Discussion: please clarify since usually when a technique is efficient we observe a rise in the use of it year by year. In the next years a rise in number of MT may be observed in Taiwan. COVID may reduce the expected slope rise and slow down the diffusion of the technique. The number of potential candidates to MT are usually below 5-10 % of ischemic strokes admitted and we have no data on the total number of stroke admitted hospitals participating to this study. The number of MT performed in the study is consistent with the potential number of candidates ? there is still a gap to fill ? Readers would be interested to know also this data.

Response 9: We very appreciated the valuable comment from the reviewer.

We had total 534 cases of patients of acute ischemic stroke admitted to our hospital. They are all received the whole procedure of evaluation of acute ischemic stroke protocal. Due to retrospective nature, we only collected the patients with COVID-19(-) who received MT for acute ischemic stroke with LVO in this study. As reviewer suggestion, we added this bias to the limitation of discussions section(line284). Still, we will make it discussed for further investigation

Reviewer 2 Report

COVID-19 can be one of the interesting topics for readers of this Journal. The main focus of this paper was on time delay which has a medium level of importance but Ischemic Stroke can be of great importance. The authors compared some variables such as door-to-puncture times and CT time, TICI Score, and mRS between two groups: pre- and during- COVID. It was better to follow up a group of patients from pre-covid to during covid to have more reliable results since the sample size of each group is not good enough. They reported that there was no difference between the pre-pandemic and during-pandemic time periods when using mechanical thrombectomy to treat COVID-19. They also showed that by means of a quick-PCR test during triage, there was no time delay to perform MT or any lowering of safety protocol for workers in the healthcare system. The main title is the Quality of Mechanical Thrombectomy but they did not discuss enough the details of the method and other related concerns of the Thrombectomy. The discussion section is well-designed and described but needs some corrections specifically to connect the present findings and the hypothesis of this study. I prefer to have the responses to the below comments from the authors.

1.       Please clarify that you follow-up some patients before and COVID. Or the patients in pre and during COVID groups were different? The abstract section about this point is not clear enough.

2.       “How to maintain effective treatment standards has become our concern. COVID-19 (-) patient …”. The abstract is not clear. Eventually, your main intention is to work on “time delays of reperfusion therapy” or the above sentence. They are different things.

3.       “COVID-19 is a potential zoonotic disease with a low to moderate (estimated 53 2%-5%) mortality rat”. It needs a reference.

4.       “… pandemic on March 19th, 2021 due to COVID-19 spreading from the national airport 64 to the general population[6, 7].” I do not know how much it is necessary to refer to a non-English-language website in a professional scientific paper. The content of these websites will change later.

5.       “Acute ischemia stroke is a significant cause of morbidity and mortality worldwide”. There are a lot of disorders in the whole world. How did you find this is “a significant” cause …

6.       “Mechanical thrombectomy (MT) has established its role as a form of first-line treatment 67 for acute ischemic stroke due to large vessel occlusions (LVOs).” It needs a reference.

7.       The last paragraph of the Introduction section is not clear enough. I can understand what was your hypothesis. But you should totally clarify your hypothesis and the parameters that you want to test your hypothesis for a better understanding of the readers.

8.       In the abstract you wrote “were selected from our single-center prospective registry” and in the first line of the Method section you wrote, “In this retrospective, single-center study”. Please clarify the type of this study.

9.       “…: 1) acute ischemic …” How did you diagnose whether are they acute or not?

10.   “ … time from onset to….” What does “onset” mean? triage?

11.   Use a capital letter for the first letters in Table 1.

12.   Can you please more explain in Fig.3. The main target of this figure is not clear to me. Even in the main manuscript, I could not find exactly what did you find from this figure.

13.   Maybe the better location for Fig. 5 can be in the Method section. But this is only a suggestion for the authors.

14.   In the first step of the Discussion section, you should compare some of your findings with previous published papers. I can understand the quantitative comparison of your results with previous results is not very easy and maybe this is not totally matched with the content of this paper. But you had some numerical data in the last paragraph of the discussion section to describe your limitations which may also be useful for this point.

15.   The details of the Mechanical Thrombectomy are not well-described and this effect on the finding of this study is not discussed. Previous studies showed many effective parameters for the quality of Thrombectomy that the authors did not discuss on them. Or the authors can discuss it as a limitation of this study. Since previous studies reported some concerns about the effect of the quality of this method on the blood hemodynamic changes that can be effective on blood clotting. Changes in blood hemodynamics can be of great importance in the Thrombectomy study and it should be discussed [https://doi.org/10.1186/s13104-019-4381-2]. In addition, another limitation of this study that can be discussed is the comorbidity effects. Some CNS disorders such as cerebral atrophy and hydrocephalus can be effective in the results that they should be discussed in this study [https://doi.org/10.3389/fbioe.2022.900644].

16.   The discussion section is well-designed and described. But the author should better connect the main results of this study about time delays and the Quality of Mechanical Thrombectomy…. The present discussion section cannot well-support the hypothesis of this study and you should more discuss connecting time delay with the title of this paper. Your title and consequently hypothesis are very general and they can include a large area.

Author Response

Response to Reviewer 2 Comments

Point 1: Please clarify that you follow-up some patients before and COVID. Or the patients in pre and during COVID groups were different? The abstract section about this point is not clear enough.

Response 1: As the reviewers recommendation, We used retrospective, single-center study to select COVID-19 (-) patients with acute ischemic stroke undergoing mechanical thrombectomy during the years 2020–2021. The times when performing reperfusion therapy were compared between patients admitted from March 2019 to February 2020 (pre- COVID-19 group) and those from March 2020 to February 2021(during COVID-19 group). Taiwan’s government did eventually announce a lockdown, the third emergency alert of the pandemic on March 19th, 2021 due to COVID-19 spreading from the national airport to the general population.

Point 2: “How to maintain effective treatment standards has become our concern. COVID-19 (-) patient …”. The abstract is not clear. Eventually, your main intention is to work on “time delays of reperfusion therapy” or the above sentence. They are different things.

Response 2: As the reviewers recommendation, In our study, with a prepared protocol for the pandemic having been established in the healthcare system, we could see no difference between the pre-pandemic and during pandemic time periods when using mechanical thrombectomy to treat COVID-19 (-) patients of AIS with LVO. By means of a quick-PCR test during triage, there was no time delay to perform MT or any lowering of safety protocol for workers in the healthcare system.

Point 3: “COVID-19 is a potential zoonotic disease with a low to moderate (estimated 53 2%-5%) mortality rat”. It needs a reference.

Response 3: As the reviewers recommendation, we have added the reference(line 53).

Ref: Wu, Y. C., Chen, C. S., & Chan, Y. J. (2020). The outbreak of COVID-19: An overview. Journal of the ChineseMedical Association : JCMA83(3), 217–220. https://doi.org/10.1097/JCMA.0000000000000270

Point 4:  “… pandemic on March 19th, 2021 due to COVID-19 spreading from the national airport 64 to the general population[6, 7].” I do not know how much it is necessary to refer to a non-English-language website in a professional scientific paper. The content of these websites will change later.

Response 4: As the reviewers recommendation, we added the website to explain the time of lockdown in Tainwan. This is an official wetsite of Taiwan Centers for Disease Control. It has Chinese/English bilingual version.

Point 5:  “Acute ischemia stroke is a significant cause of morbidity and mortality worldwide”. There are a lot of disorders in the whole world. How did you find this is “a significant” cause …

Response 5: As the reviewers recommendation, we changed the sentence with ” Acute ischemia stroke is one of the significant causes of morbidity and mortality worldwide(line 66).”

Point 6: “Mechanical thrombectomy (MT) has established its role as a form of first-line treatment 67 for acute ischemic stroke due to large vessel occlusions (LVOs).” It needs a reference

Response 6: As the reviewers recommendation, We deleted the sentence(line67)”Mechanical thrombectomy (MT) has established its role as a form of first-line treatment for acute ischemic stroke due to large vessel occlusions (LVOs)”. We added the new sentence (line68)”Mechanical thrombectomy(MT) has established its role as the standard of treatment in patients of acute ischemic stroke with large vessel occlusions(LVOs), depending on appropriate patient selection and timely reperfusion.

And, we add reference 37. (Turc G, Bhogal P, Fischer U, et al. European Stroke Organisation (ESO) – European Society for Minimally Invasive Neurological Therapy (ESMINT) Guidelines on Mechanical Thrombectomy in Acute Ischaemic StrokeEndorsed by Stroke Alliance for Europe (SAFE). European Stroke Journal. 2019;4(1):6-12. doi:10.1177/2396987319832140)

Point 7: The last paragraph of the Introduction section is not clear enough. I can understand what was your hypothesis. But you should totally clarify your hypothesis and the parameters that you want to test your hypothesis for a better understanding of the readers.

Response 7: As the reviewers recommendation, we explained the parameters in line 88.

Point 8: In the abstract you wrote “were selected from our single-center prospective registry” and in the first line of the Method section you wrote, “In this retrospective, single-center study”. Please clarify the type of this study.

Response 8:  As the reviewers recommendation, we corrected it(line31) as below: We used retrospective, single-center study to select COVID-19 (-) patients with acute ischemic stroke undergoing mechanical thrombectomy during the years 2020–2021.

Point 9:  “…: 1) acute ischemic …” How did you diagnose whether are they acute or not?performed in the study is consistent with the potential number of candidates ? there is still a gap to fill ? Readers would be interested to know also this data.

Response 9:

We had total 534 cases of patients of acute ischemic stroke admitted to our hospital. They are all received the whole procedure of evaluation of acute ischemic stroke protocal. Due to retrospective nature, we only collected the patients with COVID-19(-) who received MT for acute ischemic stroke with LVO in this study. As reviewer suggestion, we added this bias to the limitation of discussions section(line284). Still, we will make it discussed for further investigation

Point 10: “ … time from onset to….” What does “onset” mean? triage?

Response 10: As the reviewers recommendation, (line98) we have corrected the mention”time from “ symptom ” onset to ……”.

Onset : means symptom of acute ischemic stroke onset. 

Triage: Triage originates from the French word "trier," which is used to describe the processes of sorting and organization. Triage is utilized in the healthcare community to categorize patients based on the severity of their injuries and, by extension, the order in which multiple patients require care and monitoring. The history of the emergency triage originated in the military for field doctors. As early as the 18th century, documentation shows how field surgeons would quickly look over soldiers and determine if there was anything they could do for the wounded soldier. French military surgeon Baron Dominique Jean Larrey, the chief surgeon in Napoleon Bonaparte's imperial guard, developed a system based on the need to evaluate and categorize wounded soldiers quickly during battle. The triage system was first implemented in hospitals in 1964 when Weinerman et al. published a systematic interpretation of civilian emergency departments using triage. Today, triage is still deeply integrated into healthcare. Triage can be broken down into three phases: prehospital triage, triage at the scene of the event, and triage upon arrival to the emergency department. There are various triage systems implemented around the world, but the universal goal of triage is to supply effective and prioritized care to patients while optimizing resource usage and timing.

Reference:

  1. Robertson-Steel I. Evolution of triage systems. Emerg Med J. 2006 Feb;23(2):154-5. 
  2. Crumplin MK. The Myles Gibson military lecture: surgery in the Napoleonic Wars. J R Coll Surg Edinb. 2002 Jun;47(3):566-78. 
  3. Iserson KV, Moskop JC. Triage in medicine, part I: Concept, history, and types. Ann Emerg Med. 2007 Mar;49(3):275-81.

Point 11: Use a capital letter for the first letters in Table 1.

Response 11: As the reviewers recommendation, we have corrected it

Point 12: Can you please more explain in Fig.3. The main target of this figure is not clear to me. Even in the main manuscript, I could not find exactly what did you find from this figure.

Response 12: As the reviewers recommendation, we have more explantation it(line 179, line 182).

Point 13: Maybe the better location for Fig. 5 can be in the Method section. But this is only a suggestion for the authors.

Response 13: As the reviewers recommendation, we very agreed that the organization of the model should be moved to the method section (line 132).

Point 14: In the first step of the Discussion section, you should compare some of your findings with previous published papers. I can understand the quantitative comparison of your results with previous results is not very easy and maybe this is not totally matched with the content of this paper. But you had some numerical data in the last paragraph of the discussion section to describe your limitations which may also be useful for this point.

Response 14: As the reviewers recommendation, we have discussed the issue in line255 and 284.

Point 15: The details of the Mechanical Thrombectomy are not well-described and this effect on the finding of this study is not discussed. Previous studies showed many effective parameters for the quality of Thrombectomy that the authors did not discuss on them. Or the authors can discuss it as a limitation of this study. Since previous studies reported some concerns about the effect of the quality of this method on the blood hemodynamic changes that can be effective on blood clotting. Changes in blood hemodynamics can be of great importance in the Thrombectomy study and it should be discussed [https://doi.org/10.1186/s13104-019-4381-2]. In addition, another limitation of this study that can be discussed is the comorbidity effects. Some CNS disorders such as cerebral atrophy and hydrocephalus can be effective in the results that they should be discussed in this study [https://doi.org/10.3389/fbioe.2022.900644].

Response 15: As the reviewers recommendation, we have added it to the method(line 151).

Point 16:  The discussion section is well-designed and described. But the author should better connect the main results of this study about time delays and the Quality of Mechanical Thrombectomy…. The present discussion section cannot well-support the hypothesis of this study and you should more discuss connecting time delay with the title of this paper. Your title and consequently hypothesis are very general and they can include a large area.

Response 16: As the reviewers recommendation, we added the issue in line 236 and line 273.

Round 2

Reviewer 2 Report

Accept